

# Identifying biases of the Majorana scattering invariant

Isidora Araya Day[1,2*], Antonio L. R. Manesco[2],
Michael Wimmer[1,2] and Anton R. Akhmerov[2†]

**1** QuTech, Delft University of Technology, Delft 2600 GA, The Netherlands
**2** Kavli Institute of Nanoscience, Delft University of Technology,
P.O. Box 4056, 2600 GA Delft, The Netherlands

★ iarayaday@gmail.com , † detr@antonakhmerov.org

## Abstract

**The easily accessible experimental signatures of Majorana modes are ambiguous and only probe topology indirectly: for example, quasi-Majorana states mimic most properties of Majoranas. Establishing a correspondence between an experiment and a theoretical model known to be topological resolves this ambiguity. Here we demonstrate that already theoretically determining whether a finite system is topological is by itself ambiguous. In particular, we show that the scattering topological invariant—a probe of topology most closely related to transport signatures of Majoranas—has multiple biases in finite systems. For example, we identify that quasi-Majorana states also mimic the scattering invariant of Majorana zero modes in intermediate-sized systems. We expect that the bias due to finite size effects is universal, and advocate that the analysis of topology in finite systems should be accompanied by a comparison with the thermodynamic limit. Our results are directly relevant to the applications of the topological gap protocol.**

## 1 Introduction

The quest to create a topological phase inevitably faces an obstacle: how to determine whether a system is in fact topological? Unlike simulations that reveal all the information about a system, experimental probes are limited and do not directly measure the signatures of topology [1–4]. Furthermore, even defining what topological means in a finite system, rather than in the thermodynamic limit, is inherently ambiguous. The celebrated Pfaffian invariant $\text{sign}(\text{Pf}[iH(k=0)]\text{Pf}[iH(k=\pi)])$, for example, tells whether a sufficiently long one-dimensional superconductor hosts Majorana zero modes at its boundaries [5]. Finite size effects couple the Majorana zero modes and give them an energy splitting, which is exponentially small in the length of the superconductor. Therefore, the question of whether a finite sample is topological is as ambiguous as asking whether an exponentially small energy splitting is zero.

An approach to determine the presence of Majorana zero modes in small systems is the scattering invariant [3,6]. To compute it, we attach a metallic lead to both ends of a Majorana

nanowire and obtain the reflection matrix $r$ that relates the incoming and outgoing wave functions from a lead. In the presence of Majorana zero modes, $\text{sign}\det r = -1$, while in their absence, $\text{sign}\det r = 1$ [3, 6, 7]. Because the determinant of the reflection matrix may only change sign upon the appearance of transmitting modes along the nanowire, phase transitions in the scattering invariant are directly related to a closure of the transport gap. Away from the phase transition the nontrivial scattering invariant predicts the appearance of the zero bias local conductance peak—an experimental signature of Majorana zero modes [1]. This approach has been used to compute the topological invariant in disordered nanowires [8, 9], because it does not require translational invariance and it is computationally efficient.

It is well known that in a finite superconductor the scattering invariant turns trivial if the leads are coupled weakly to the Majorana zero modes by tunnel barriers [3]. This effect vanishes as the length of the nanowire becomes larger, with the scattering invariant converging to topological in the thermodynamic limit. This bias towards the invariant being trivial complicates finding parameters that realize a topological phase. A reverse bias is much more dangerous: identifying a small system as topological while a longer one would be trivial may lead research in an incorrect direction. To mitigate this risk, we answer the following question: what are the mechanisms that lead to a biased interpretation of the scattering invariant in Majorana nanowire simulations? In particular, we investigate biases of the scattering invariant in the presence of quasi-Majorana modes—zero energy modes that are not topologically protected [10–12].

## 2 Scattering invariant in the strongly coupled limit

The first step to compute a scattering invariant is to define a quantum transport setup where metallic leads are attached to a scattering region. The scattering matrix $S$ relates the amplitudes of the incoming and outgoing modes in the leads:

$$q_{\text{out}} = S q_{\text{in}}\,, \tag{1}$$

where $q_{\text{in}}$ and $q_{\text{out}}$ are vectors with the modes amplitudes. A direct way to compute the scattering matrix is to use the Hamiltonian of the entire system and the semi-infinite leads and solve the scattering equations numerically [13], for example using the Kwant package [14]. Alternatively, in the weak coupling limit, the Mahaux-Weidenmüller formula [15] provides an approximation of the scattering matrix:

$$S(E) = 1 - 2\pi i W \left(E - H + i\pi W^\dagger W\right)^{-1} W^\dagger\,, \tag{2}$$

where $H$ is the low-energy Hamiltonian of the scattering region, $E$ is the energy of the incoming modes, and $W$ is the coupling between the lead and the low-energy states of the scattering region. Because $H$ and $W$ only contain low-energy degrees of freedom, the matrices are small, making the Mahaux-Weidenmüller formula especially useful to compute the scattering matrix analytically.

The single metal-superconductor interface shown in Fig. 1(a) is sufficient to define the scattering invariant in the thermodynamic limit. As long as the superconductor is gapped, all the sub-gap electron and hole modes that approach the superconductor reflect back into the metallic lead, such that the scattering matrix only consists of a reflection matrix, $S = r$. Because a Hermitian system conserves the total particle number, $S$ is unitary, making $r^\dagger r = 1$. In the particle-hole basis, the reflection matrix is a $2 \times 2$ block-matrix that relates the incoming and outgoing electron and hole modes:

$$r = \begin{pmatrix} r_{ee} & r_{eh} \\ r_{he} & r_{hh} \end{pmatrix}\,, \tag{3}$$

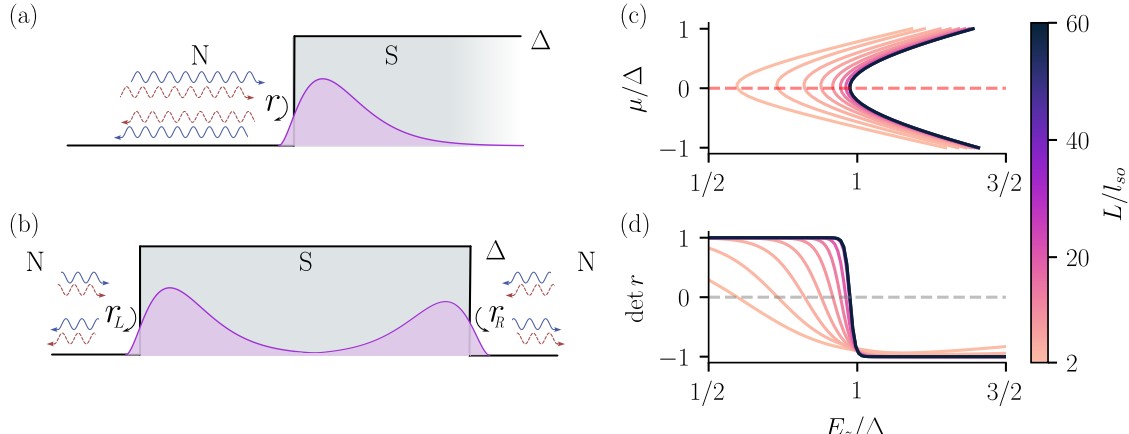

Figure 1: Quantum transport setup for computing the scattering invariant in a Majorana nanowire. (a) Normal metallic lead (white) attached to a superconducting nanowire (gray), a one-terminal setup. The incoming electron (blue) and hole (red) modes reflect back from the gapped superconductor. The reflection matrix $r$ encodes the presence of Majorana zero modes (purple). (b) Two-terminal setup with a metallic lead attached to each end of the nanowire. In the thermodynamic limit, the incoming electron and hole modes from both leads reflect back from the superconductor. (c) Phase diagram for nanowires of different lengths, with $l_{so} = t_a/\alpha$ the spin-orbit length and $a$ the lattice constant. The lines show the parameter values where the scattering invariant changes sign. (d) Determinant of the reflection matrix as a function of the Zeeman field across the phase transition for fixed $\mu = 0$, as shown by the red dashed line in (c). Details of the simulation are in the appendix.

where particle-hole symmetry constraints ensure that $r_{ee}(E) = r_{hh}^*(-E)$ and $r_{eh}(E) = r_{he}^*(-E)$. The combination of both constraints at $E = 0$ restricts

$$\mathcal{Q} = \det r \,, \tag{4}$$

to only take values $\mathcal{Q} = \pm 1$. To demonstrate that $\mathcal{Q}$ is a topological invariant, we use the Mahaux-Weidenmüller formula in Eq. (2) to compute the scattering matrix with a minimal Hamiltonian in both limits. The trivial limit has no sub-gap modes, therefore $S = 1$ and $\mathcal{Q} = 1$. The topological limit has a single Majorana zero mode at the interface, so that $H = 0$ and $W = (t_L, t_L^*)^T$ where we choose the coupling to the lead $t_L$ to be real. This gives $S(E = 0) = -\sigma_x$ and $\mathcal{Q} = -1$. Furthermore, the value of $\mathcal{Q}$ may only change if the superconducting gap closes and electrons and holes transmit into the superconductor [3,6], a feature directly related to the appearance of a peak in non-local conductance measurements [3,4]. The gap closing points separate the trivial regions with $\mathcal{Q} = 1$ from the topological regions with $\mathcal{Q} = -1$.

Counterintuitively, a simple argument shows that making the superconductor finite always gives a trivial scattering invariant. Let us consider a metallic lead attached to a finite trivial region which we gradually tune into a topological phase. For the finite region to undergo a topological phase transition, $\det r$ must continuously change sign and thus cross zero. This is however impossible if there is only one lead attached to the system, because for $r$ to have a zero eigenvalue a transmission into another lead must appear. As a consequence, the invariant cannot change sign and remains $\mathcal{Q} = 1$ for all parameter values. This apparent contradiction appears due to the resonant coupling between the lead and the Majorana zero mode at the terminated end of the superconductor. Therefore, to compute the scattering invariant in a finite system, we must attach two leads, as shown in Fig. 1(b). In this case, the scattering

matrix is a $2 \times 2$ block-matrix that relates the incoming and outgoing modes in the left (L) and right (R) leads:

$$S = \begin{pmatrix} r_L & t_{LR} \\ t_{RL} & r_R \end{pmatrix}, \tag{5}$$

where the blocks $r_L$ and $r_R$ are the reflection matrices for each lead, and $t_{LR}$ and $t_{RL}$ are the transmission matrices between the leads. Because $S$ is unitary, $r_L$ and $r_R$ are sub-unitary instead, and may have zero eigenvalues, which in turn correspond to the phase transition in a finite system. In a two-terminal setup we use $\mathcal{Q} = \text{sign} \det r_L = \text{sign} \det r_R$. Particle conservation and particle-hole symmetry constraints ensure that the scattering invariant is the same in both leads in a two-terminal setup [3], a result we also confirmed numerically throughout this work.

To compare the scattering invariant in the thermodynamic limit and finite systems, we simulate a microscopic one-dimensional nanowire [16, 17] using the Kwant package [14]. The nanowire has a chemical potential $\mu$, hopping $t$, Zeeman field $E_Z$, lattice spin-orbit coupling $\alpha$, and superconducting pairing $\Delta$, and the leads are modeled by setting $\Delta = 0$. Details of the simulation are in the appendix and the code for this figure and the rest of the paper are available in Ref. [18]. In the thermodynamic limit, the phase transition occurs at $E_Z = \sqrt{\mu^2 + \Delta^2}$. Figure 1(c) shows the phase transition of finite nanowires of different lengths, which we determine by finding the parameters for which the scattering invariant changes sign. This simulation demonstrates the first bias when interpreting the scattering invariant in finite systems: if a nanowire is not sufficiently long, the transition to a topological phase may appear to be at a smaller critical field than in the thermodynamic limit. This is a counterintuitive result, because shorter nanowires are expected to have a larger energy splitting between the Majorana zero modes, and therefore a larger critical field. We observe that changing the chemical potential in the leads shifts the phase transition to larger Zeeman fields, indicating that the scattering invariant is sensitive to the self-energy of the leads. We thus attribute the bias to the self-energy of the leads: the finite Zeeman field splits the electron and hole modes in the leads, which has a back-action in the properties of the reflection matrix close to the phase transition. Despite the bias, Fig. 1(d) shows that for Zeeman fields lower than the true critical value the transmission between the two leads stays sizeable. The quantity $\det r$ is also known in the literature as the topological visibility [19–21].

## 3 Scattering invariant in the tunneling limit

The back-action of the lead on the scattering region becomes smaller if the lead is coupled through a tunnel barrier. Tunnel barriers are also useful to identify individual states through resonant tunneling and they have practical advantages for measuring non-local conductance. Because we have identified the self-energy of the leads as a source of bias, it is natural to consider tunnel barriers as a solution to this problem. In this section we show that tunnel barriers introduce their own biases too.

### 3.1 Strong Majorana overlap

An effective description of the finite nanowire with tunnel barriers is given by the Hamiltonian:

$$H_{\text{NW}} = \begin{pmatrix} 0 & iE_M \\ -iE_M & 0 \end{pmatrix}, \qquad W = \begin{pmatrix} t_L & 0 \\ t_L^* & 0 \\ 0 & t_R \\ 0 & t_R^* \end{pmatrix}, \tag{6}$$

where $H_{\mathrm{NW}}$ is the Hamiltonian in the Majorana basis and $W$ is the coupling matrix between the Majorana zero modes and the leads. The columns of $W$ are in the Majorana basis, while the rows are in the electron and hole mode basis of the right and left leads, $\{\psi_{L,e}, \psi_{L,h}, \psi_{R,e}, \psi_{R,h}\}$. The coupling $E_M$ between the Majorana zero modes is exponentially small in the length of the nanowire, and the tunnel barriers determine the tunneling amplitudes $t_L$ and $t_R$ between the left and right leads and the Majorana zero modes, respectively. Here we disregard the coupling between the Majorana zero modes and the lead at the opposite end of the nanowire for simplicity.

To find an analytical expression for the scattering invariant, we substitute Eq. (6) into the Mahaux-Weidenmüller formula (2):

$$\det r = \frac{E_M^2 - \Gamma_L \Gamma_R}{E_M^2 + \Gamma_L \Gamma_R} = \begin{cases} < 0, & \text{if } E_M < \sqrt{\Gamma_L \Gamma_R}, \\ > 0, & \text{if } E_M > \sqrt{\Gamma_L \Gamma_R}, \end{cases} \tag{7}$$

where $\Gamma_i = 2\pi |t_i|^2$. This result constitutes another bias: the scattering invariant is agnostic to the presence of Majorana zero modes if the coupling to the leads is smaller than the Majoranas' energy splitting [3], $\Gamma_i \ll E_M$. We also confirm this bias beyond the weak coupling limit by solving the scattering equations numerically in a microscopic nanowire with two Gaussian-shaped tunnel barriers of height $V_0$, as shown in Fig. 2(a). Because $E_M$ is exponentially small in the length of the nanowire, the scattering invariant may indicate a trivial phase in a system that is topological in the thermodynamic limit.

## 3.2 Quasi-Majorana strong overlap

That the scattering invariant is blind to modes that are weakly coupled to the leads is no surprise: in the limit where a mode is not coupled at all, it cannot be detected. The mechanism, however, raises an interesting question: are there any regimes where trivial states may be misinterpreted as topological? Generally, two trivial bound states localized at the same end of the nanowire couple to each other and gap out. However, the presence of a smooth position-dependent potential may suppress the hybridization of the bound states, making them robust to changes in the systems parameters [10–12]. Due to their stability and the similarity of their signatures to Majorana zero modes [22,23], these states are known as quasi-Majorana modes. Distinguishing them is an open challenge in the field.

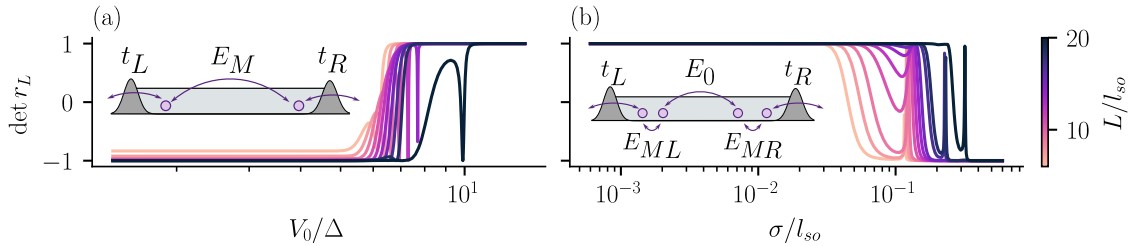

Figure 2: Scattering invariant of a finite nanowire with symmetric tunnel barriers. (a) Nanowire in the Majorana regime, $\mu^2 + \Delta^2 > E_z^2$, with varying tunnel barrier amplitude $V_0$. (b) Nanowire in the quasi-Majorana regime (trivial) as a function of the tunnel barrier width $\sigma$. The insets illustrate the low-energy degrees of freedom (purple circles) with their effective couplings (arrows).

The effective Hamiltonian of a nanowire with quasi-Majorana modes is:

$$H_{\text{NW}} = \begin{pmatrix} 0 & iE_{ML} & 0 & 0 \\ -iE_{ML} & 0 & iE_0 & 0 \\ 0 & -iE_0 & 0 & iE_{MR} \\ 0 & 0 & -iE_{MR} & 0 \end{pmatrix}, \quad W = \begin{pmatrix} t_L & 0 & 0 & 0 \\ t_L^* & 0 & 0 & 0 \\ 0 & 0 & 0 & t_R \\ 0 & 0 & 0 & t_R^* \end{pmatrix}, \tag{8}$$

where $H_{\text{NW}}$ and the columns of $W$ are in the Majorana basis that label the four quasi-Majorana modes, while the rows are the same as in the previous case. $E_{ML}$ and $E_{MR}$ couple quasi-Majorana modes at the same end of the nanowire, while $E_0$ couples the quasi-Majorana modes at opposite ends. For simplicity, we only consider nearest-neighbor couplings between the quasi-Majorana modes and the leads, as shown in the inset of Fig. 2(b). Once again we use the Mahaux-Weidenmüller formula (2) to obtain an analytical expression for the scattering invariant:

$$\det r = \frac{-E_0^2 \Gamma_L \Gamma_R + E_{ML}^2 E_{MR}^2}{E_0^2 \Gamma_L \Gamma_R + E_{ML}^2 E_{MR}^2}, \tag{9}$$

where $\Gamma_i = 2\pi |t_i|^2$. Remarkably, in the regime where one pair of quasi-Majorana modes is strongly coupled to the leads while the other is not, $|\Gamma_L|, |\Gamma_R|, E_0 \gg E_{ML}, E_{MR}$, the scattering invariant becomes $\mathcal{Q} = -1$. This is shown in Fig. 2(b) for a microscopic one-dimensional nanowire with Gaussian-shaped tunnel barriers, where the width $\sigma$ of the tunnel barriers controls $E_{ML}$, $E_{MR}$, $\Gamma_L$, and $\Gamma_R$. Without further analysis, one may incorrectly interpret the trivial quasi-Majorana modes as Majorana zero modes.

## 3.3 Quasiparticle sinks

That the scattering invariant may be computed from the left or right leads in a two-terminal setup is a general and robust property that holds for any $2 \times 2$ block-matrix scattering matrix. We illustrate this using random matrices and computing the determinant of the diagonal blocks. The results are shown in Fig. 3(a): both blocks always share the same determinant, as expected in a two-terminal setup. This property breaks in the presence of additional quasiparticle sinks or sources in the system, for example an additional lead attached to the nanowire. Any other mechanism that loses particles into the environment, like superconducting vortices or non-hermitian effects, has a similar consequence. We illustrate the breakdown of the equivalence between the scattering invariants using random matrices with a $3 \times 3$ block structure in Fig. 3(b), where the third block represents the quasiparticle sink. The impact of quasiparticle sinks in the scattering invariant is relevant in interpreting the results the topological gap protocol [8,9] because the simulation results used to calibrate it [24] show in in Fig. 3(c) a significant deviation from the two-terminal behavior. This deviation likely occurs due to the presence of a Dynes parameter mentioned in Ref. [8].

We consider the specific case of a superconductor where the quasiparticles have a finite lifetime and decay into the environment. This is often modeled using an imaginary diagonal term in the superconducting Hamiltonian—the Dynes parameter—which results in a non-hermitian self-energy [19]. We study the effect of the Dynes parameter $\eta > 0$ on the scattering invariant of a nanowire with quasi-Majorana modes. We focus on the effective Hamiltonian of one end of the nanowire:

$$H_{\text{NW}} = \begin{pmatrix} -i\eta & iE_{ML} \\ -iE_{ML} & -i\eta \end{pmatrix}, \quad W = \begin{pmatrix} t_L & 0 \\ t_L^* & 0 \\ 0 & 0 \\ 0 & 0. \end{pmatrix}, \tag{10}$$

where $H_{\text{NW}}$ is the Hamiltonian in the Majorana basis and $W$ is the coupling matrix between two quasi-Majorana modes at one end of the nanowire and the corresponding lead, as in

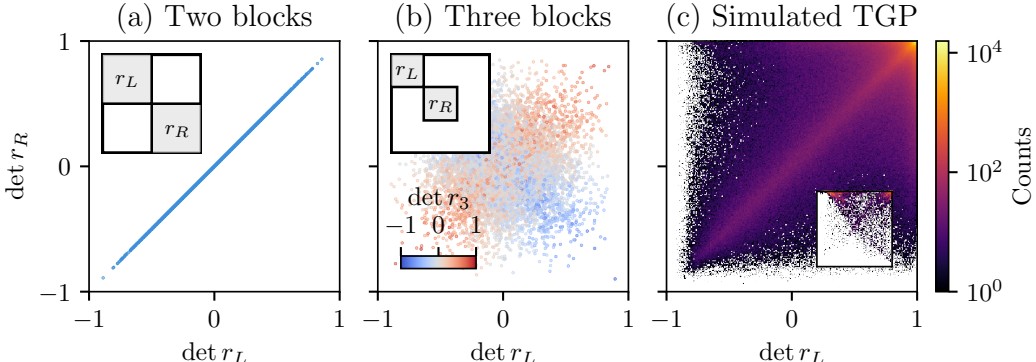

Figure 3: Distribution of the scattering invariant computed from the left and right leads in correct and incorrect setups. (a) Determinant of the diagonal blocks of $2\times2$ special orthogonal matrices sampled randomly. (b) Determinant of the diagonal blocks of $3\times3$ special orthogonal matrices sampled randomly. (c) Superimposed data of different experiments from the benchmarks of the topological gap protocol [9,24]. The inset shows the data for the simulation with the largest variance of $\det r_R - \det r_L$.

Fig. 4(a). For simplicity we disregard the coupling to the other quasi-Majorana modes. Using the Mahaux-Weidenmüller formula we find the scattering invariant:

$$\det r = \frac{E_{ML}^2 + \eta^2 - \eta\Gamma_L}{E_{ML}^2 + \eta^2 + \eta\Gamma_L}, \tag{11}$$

where $\Gamma_L = 2\pi|t_L|^2$ Equation (11) shows that the scattering invariant is topologically non-trivial in the regime $\Gamma_L \gg \eta, E_M^2/\eta$, even though the quasi-Majorana modes are trivial. We confirm this result numerically in a microscopic one-dimensional nanowire with symmetric tunnel barriers, as shown in Fig. 4(b). Furthermore, in Fig. 4(c) we show that the scattering invariant computed from the left and right leads does not agree in the presence of the Dynes parameter if the coupling to the leads is also asymmetric. This is the third bias we identify, and it demonstrates that the level broadening introduced in Ref. [8,9] may systematically make the scattering invariant topologically nontrivial in the quasi-Majorana regime, even while keeping the scattering matrix approximately unitary. Even worse than the other cases, this bias persists in the thermodynamic limit, when the two ends of the nanowire are decoupled. It therefore invalidates the topological visibility as a reliable indicator of Majoranas in a system with a single NS interface and dissipative broadening [19].

## 4 Discussion

We demonstrated multiple ways in which a scattering invariant of a finite system is biased compared to the thermodynamic limit:

- In the open regime, the back-action from the leads enhances Zeeman splitting and pushes the topological transition to smaller fields.

- Weak tunnel couplings to the leads allow Majoranas at the opposite ends of the system to couple, so that the system appears trivial.

- Similarly, resolving the coupling between the quasi-Majorana modes at different ends of the system makes a trivial system appear topological.

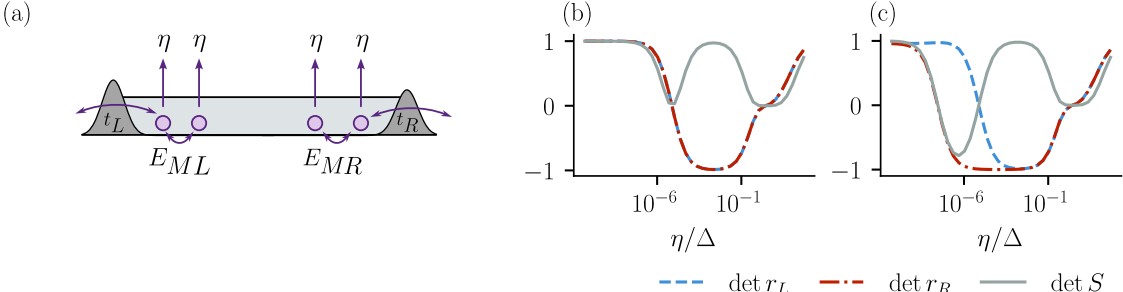

Figure 4: Nanowire with quasi-Majorana modes and a Dynes parameter. (a) Illustration of the low-energy degrees of freedom (purple circles) with their effective couplings (arrows). (b) Scattering invariant as a function of the Dynes parameter $\eta$ for a nanowire with two symmetric tunnel barriers. (c) Scattering invariant as a function of the Dynes parameter $\eta$ for a nanowire with two asymmetric tunnel barriers of different heights and widths.

These biases diminish as the system size is increased, but they are likely to be relevant to the ongoing experimental efforts. In addition to these biases, we demonstrated that quasiparticle sinks may make quasi-Majorana states appear topological also in the thermodynamic limit.

Our analysis focused on the scattering invariant because of its relation to the transport properties, however, the finite size effects unavoidably affect other topological invariants too. We therefore propose to always combine the analysis of the finite system with a comparison to the behavior in the thermodynamic limit. In disordered systems such analysis must also include disorder averaging and confirming that the results are not influenced by insufficient averaging.

## Acknowledgments

We thank Henry Legg for drawing our attention to the determinant sign mismatch in Ref. [9]. We thank Roman Lutchyn, Chetan Nayak, Dmitry Pikulin, and Andrey Antipov for useful discussions regarding the topological gap protocol. We acknowledge fruitful discussions with Kostas Vilkelis and Binayyak Bhusan Roy.

**Data availability**  The code used to produce the reported results and the generated data are available on Zenodo [18].

**Author contributions**  A. A. and M. W. proposed the research idea. All authors defined the project scope. I. A. D. developed the simulations with input from all authors. I. A. D. and A. M. produced the figures with input from M. W. and A. A. Finally, I. A. D, A. M. and A. A. wrote the manuscript with input from M. W.

**Funding information**  This work was supported by the Netherlands Organization for Scientific Research (NWO/OCW) as part of the Frontiers of Nanoscience program and OCENW.GROOT.2019.004.

# A   Details of the tight-binding model

Throughout this work, we perform the numerical simulations using a one-dimensional model of a semiconducting nanowire proximitized by a superconductor [16, 17]. The tight-binding Hamiltonian is given by:

$$H = \sum_n \Psi_n^\dagger \mathcal{H} \Psi_n + \Psi_n^\dagger \mathcal{H}^{hop} \Psi_{n+1} + \text{h.c.},$$

$$\mathcal{H} = (2t - \mu)\tau_z + \Delta\tau_x + E_Z\sigma_z, \quad \mathcal{H}^{hop} = \left(-t + \frac{i\alpha}{2}\sigma_y\right)\tau_z, \tag{A.1}$$

where $\Psi = (c_{n,\uparrow}, c_{n,\downarrow}, -c_{n,\downarrow}^\dagger, c_{n,\uparrow}^\dagger)^T$ is the Nambu spinor of the annihilation operators $c_{n,\sigma}$ of electrons with spin $\sigma$ at site $n$. The Pauli matrices $\tau_i$ and $\sigma_i$ act on the particle-hole and spin degrees of freedom, respectively. The hopping amplitude between nearest neighbors is $t$, which we set to 1, $\mu$ is the chemical potential, $\alpha$ is the Rashba spin-orbit coupling strength, $E_Z$ is the Zeeman energy parallel to the wire. The superconducting pairing potential $\Delta$ is finite in the nanowire, and absent in the normal leads.

Additionally, to demonstrate the biases in Fig. 2 and Fig. 4 we consider tunnel barriers at the ends of the nanowire, which modulate the coupling to the leads. We model the tunnel barriers as a Gaussian potential:

$$H_{\text{barrier}} = \sum_{l=L,R}\sum_n \Psi_n^\dagger \left\{ V_l \exp\left[-\frac{(x_n - x_0)^2}{2\sigma_l^2}\right]\tau_z \right\}\Psi_n, \tag{A.2}$$

where $x_{l=L,R}$ are the center positions of the tunnel barriers, $V_{l=L,R}$ are their maximal heights, and $\sigma_{l=L,R}$ are their standard deviations.

We implement the tight-binding Hamiltonian with the Kwant package [14] and use it to obtain the scattering matrix at zero energy. To ensure that the scattering matrix is real, we provide the particle-hole operator $\mathcal{P} = \sigma_y\tau_y$ to Kwant, see Ref. [18] for the code. To produce Fig. 1(c-d) in the main text, we set $\Delta = 0.01$ and $\alpha = 0.1$ in a nanowire with $L = 600$ sites. In Fig. 2(a) we use $\mu = 0.1$, $\Delta = 0.05$, $\alpha = 0.02$, $B = 0.2$, $\sigma_L = \sigma_R = 10$ to ensure the Majorana regime in a finite nanowire with $L = 1000$ sites. In Fig. 2(b) we use $\mu = 0.12$, $\Delta = 0.05$, $\alpha = 0.02$, $E_Z = 0.1$, and $V_L = V_R = 0.15$ to ensure the quasi-Majorana regime in a finite nanowire with $L = 1000$ sites. We define $l_{\text{so}} = t_a/\alpha$ as the spin-orbit length, where $a$ the lattice constant.

Finally, to demonstrate the effects of quasiparticle loss, we add a Dynes parameter $\eta$ to the nanowire Hamiltonian:

$$H_{\text{loss}} = -i\sum_n \eta\Psi_n^\dagger\Psi_n. \tag{A.3}$$

This is a minimal model for a non-hermitian self-energy term in a superconducting system. Figure 4 is computed for $\eta \in [10^{-12}, 10]$, $\mu = 0.12$, $\Delta = 0.02$, $\alpha = 0.2$, $E_Z = 0.1$, $V_L = V_R = 0.1$, and $L = 1000$ sites. In Fig. 4(a) we use $\sigma_L = \sigma_R = 10$, and in Fig. 4(b) we use $\sigma_L = 10$ and $\sigma_R = 20$.

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
