# Peer review of "Identifying biases of the Majorana scattering invariant"

_SciPost Physics, doi:SciPost Phys. Core 8, 047 (2025)_

## Round 1 · Referee Report · Anonymous (Referee 1) · 2025-5-20

Strengths
1- The manuscript is very well-written. It has beautiful figures and exceedingly clear text with no unnecessarily complicated language. 2- There is a clear, well-supported message about a model system with implications that are generally relevant to topological insulators and superconductors. 3- The manuscript contains a powerful combination of analytical results for simple, effective model systems and precise numerical simulations of lattice models. 4- The findings are relevant to the search for Majorana zero modes using the "topological gap protocol," a large-scale research project currently underway at Microsoft.
Weaknesses
1- The community and literature are well aware that the scattering invariant and other indicators may fail for finite systems. In particular, it is known that the Majorana scattering invariant is reliable only if |det r| is close to 1.
Report
That topological invariants are ambiguous for finite-size systems is well known in the community — at least among those who have carefully considered the identification of topological phases. The inability to identify bulk topology from finite-size systems has been a recurring theme in the literature on Majorana wires since the early days of the field. See, for example, Pikulin and Nazarov, JETP Letters 94, 693 (2011). Conversely, experimental efforts to realize Majorana zero modes in hybrid semiconductor-superconductor systems feature finite-length nanowires. A thorough understanding of the pitfalls of commonly used indicators, such as the scattering invariant, is essential for properly identifying topological phases. In the "topological gap protocol" of Ref. 8, the Majorana scattering invariant appears to be used as an indicator of the topological phase without explicitly mentioning such caveats. Therefore, one could argue that part of the community would benefit from a clear exposition of why topological indicators in finite-size systems may fail and how this occurs.
All six of the general acceptance criteria for publication in SciPost are generously met. However, whether the manuscript meets the "groundbreaking discovery" criterion for publication in SciPost Physics is questionable. While not all readers may find the article's message new, I expect they will find it important and relevant to the ongoing quest for Majorana zero modes. If that suffices for publication in SciPost Physics, I enthusiastically support it. Otherwise, it will be an excellent article for SciPost.
Requested changes
The manuscript is very well-written. No major changes are needed. One small item: The variable t in the caption of Figure 3 is not defined in the main text. It should be defined somewhere in the text.
Recommendation
Publish (meets expectations and criteria for this Journal)

---

## Round 1 · Referee Report · Anonymous (Referee 2) · 2025-5-31

Strengths
2 - immediately relevant to ongoing experimental efforts to identify topological phases
3 - succinct in making its main points
4 - numerical and analytical models are kept as simple as possible to drive home each point
Weaknesses
Report
The paper makes these points very concisely and appears to be an excellent piece of research. It is very timely, adding some much needed clarity to the ongoing discussion in the context of Microsoft's topological quantum claims.
However, I would not refer to this analysis as a "groundbreaking theoretical discovery" and therefore suggest to publish this (as is) in scipost physics core instead. While these are very relevant results, I'd call this "overdue diligence".
Requested changes
No changes needed.
Recommendation
Accept in alternative Journal (see Report)

---

## Editorial Decision

published